# TUNING RECURRENT NEURAL NETWORKS WITH REINFORCEMENT LEARNING

**Natasha Jaques**[12]**, Shixiang Gu**[134]**, Richard E. Turner**[3]**, Douglas Eck**[1]

[1]Google Brain, USA
[2]Massachusetts Institute of Technology, USA
[3]University of Cambridge, UK
[4]Max Planck Institute for Intelligent Systems, Germany
`jaquesn@mit.edu, sg717@cam.ac.uk, ret26@cam.ac.uk, deck@google.com`

## ABSTRACT

The approach of training sequence models using supervised learning and next-step prediction suffers from known failure modes. For example, it is notoriously difficult to ensure multi-step generated sequences have coherent global structure. We propose a novel sequence-learning approach in which we use a pre-trained Recurrent Neural Network (RNN) to supply part of the reward value in a Reinforcement Learning (RL) model. Thus, we can refine a sequence predictor by optimizing for some imposed reward functions, while maintaining good predictive properties learned from data. We propose efficient ways to solve this by augmenting deep Q-learning with a cross-entropy reward and deriving novel off-policy methods for RNNs from KL control. We explore the usefulness of our approach in the context of music generation. An LSTM is trained on a large corpus of songs to predict the next note in a musical sequence. This *Note-RNN* is then refined using our method and rules of music theory. We show that by combining maximum likelihood (ML) and RL in this way, we can not only produce more pleasing melodies, but significantly reduce unwanted behaviors and failure modes of the RNN, while maintaining information learned from data.

## 1 INTRODUCTION

Generative modeling of music with deep neural networks is typically accomplished by training a RNN such as a Long Short-Term Memory (LSTM) network to predict the next note in a musical sequence (e.g. Eck & Schmidhuber (2002)). Similar to a *Character RNN* (Mikolov et al., 2010), these *Note RNNs* can be used to generate novel melodies by initializing them with a short sequence of notes, then repeatedly sampling from the model's output distribution generated to obtain the next note. While melodies and text generated in this way have recently garnered attention[1], this type of model tends to suffer from common failure modes, such as excessively repeating tokens, or producing sequences that lack a consistent theme or structure. Such sequences can appear wandering and random (see Graves (2013) for a text example).

Music compositions adhere to relatively well-defined structural rules, making music an interesting sequence generation challenge. For example, music theory tells that groups of notes belong to keys, chords follow progressions, and songs have consistent structures made up of musical phrases. Our research question is therefore whether such music-theory-based constraints can be learned by an RNN, while still allowing it to maintain note probabilities learned from data.

To approach this problem we propose *RL Tuner*, a novel sequence learning approach in which RL is used to impose structure on an RNN trained on data. The reward function in our framework combines task-related rewards with the probability of a given action originally learned by the pre-trained RNN. Thus, our model directly preserves inforamtion about the original probability distributions learned from data, while allowing us to explicitly control the trade-off between the influence of data

---

[1]http://www.theverge.com/2016/6/1/11829678/google-magenta-melody-art-generative-artificial-intelligence

and heuristic rewards. This is an important novel direction of research, because in many tasks the available reward functions are not a perfect metric that alone will lead to the best task performance in the real world (e.g. BLEU score). Unlike previous work (e.g. (Ranzato et al., 2015), (Bahdanau et al., 2016), (Norouzi et al., 2016), (Li et al., 2016)) we do not use ML training as a way to simply bootstrap the training of an RL model, but rather we rely mainly on information learned from data, and use RL only as a way to refine characteristics of the output by imposing structural rules.

This paper contributes to the sequence training and RL literature by a) proposing a novel method for combining ML and RL training; b) showing the connection between this approach and Stochastic Optimal Control (SOC)/KL-control with a pre-trained RNN as a prior policy; c) showing the explicit relationships among a generalized variant of $\Psi$-learning (Rawlik et al., 2012), G-learning (Fox et al.), and Q-learning with log prior augmentation; d) being the first work to explore generalized $\Psi$-learning and G-learning with deep neural networks, serving as a reference for exploring KL-regularized RL objectives with deep Q-learning; e) empirically comparing generalized $\Psi$-learning, G-learning, and Q-learning with log prior augmentation for the first time; and f) applying this new technique to the problem of music generation, and showing through an empirical study that this method produces melodies which are more melodic, harmonious, interesting, and rated as significantly more subjectively pleasing, than those of the original *Note RNN*. We suggest that the *RL Tuner* method could have potential applications in a number of areas as a general way to refine existing recurrent models trained on data by imposing constraints on their behavior.

## 2 BACKGROUND

### 2.1 DEEP Q-LEARNING

In RL, an agent interacts with an environment. Given the state of the environment at time $t$, $s_t$, the agent takes an action $a_t$ according to its policy $\pi(a_t|s_t)$, receives a reward $r(s_t, a_t)$, and the environment transitions to a new state, $s_{t+1}$. The agent's goal is to maximize reward over a sequence of actions, with a discount factor of $\gamma$ applied to future rewards. The optimal deterministic policy $\pi^*$ is known to satisfy the following Bellman optimality equation,

$$Q(s_t, a_t; \pi^*) = r(s_t, a_t) + \gamma \mathbb{E}_{p(s_{t+1}|s_t, a_t)}[\max_{a_{t+1}} Q(s_{t+1}, a_{t+1}; \pi^*)] \tag{1}$$

where $Q^\pi(s_t, a_t) = \mathbb{E}_\pi[\sum_{t'=t}^\infty \gamma^{t'-t} r(s_{t'}, a_{t'})]$ is the $Q$ function of a policy $\pi$. $Q$-learning techniques (Watkins & Dayan, 1992; Sutton et al., 1999) learn this optimal Q function by iteratively minimizing the Bellman residual. The optimal policy is given by $\pi^*(a|s) = \arg\max_a Q(s, a)$. Deep $Q$-learning(Mnih et al., 2013) uses a neural network called the *deep Q-network* (DQN) to approximate the $Q$ function $Q(s, a; \theta)$. The network parameters $\theta$ are learned by applying stochastic gradient descent (SGD) updates with respect to the following loss function,

$$L(\theta) = \mathbb{E}_\beta[(r(s, a) + \gamma \max_{a'} Q(s', a'; \theta^-) - Q(s, a; \theta))^2] \tag{2}$$

where $\beta$ is the exploration policy, and $\theta^-$ is the parameters of the *Target Q-network* (Mnih et al., 2013) that is held fixed during the gradient computation. The moving average of $\theta$ is used as $\theta^-$ as proposed in (Lillicrap et al., 2016). Exploration can be performed with either the $\epsilon$-greedy method or Boltzmann sampling. Additional standard techniques such as *replay memory* (Mnih et al., 2013) and *Deep Double Q-learning* (Hasselt et al., 2015) are used to stablize and improve learning.

### 2.2 MUSIC GENERATION WITH LSTM

Previous work with music generation using deep learning (e.g. (Eck & Schmidhuber, 2002), (Sturm et al., 2016)) has involved training an RNN to learn to predict the next note in a monophonic melody; we call this type of model a *Note RNN*. Often, the *Note RNN* is implemented using a Long Short-Term Memory (LSTM) network (Gers et al., 2000). LSTMs are networks in which each recurrent cell learns to control the storage of information through the use of an input gate, output gate, and forget gate. The first two gates control whether information is able to flow into and out of the cell, and the latter controls whether or not the contents of the cell should be reset. Due to these properties, LSTMs are better at learning long-term dependencies in the data, and can adapt more rapidly to new data (Graves, 2013). A softmax function can be applied to the final outputs of the network to obtain

the probability the network places on each note, and softmax cross-entropy loss can be used to train the model via back propagation through time (BPTT) (Graves & Schmidhuber, 2005). However, as previously described, the melodies generated by this model tend to wander, and lack musical structure; we will show that they are also perceived as less musically pleasing by listeners. In the next section, we will show how to improve this model with RL.

## 3 RL TUNER DESIGN

Given a trained *Note RNN*, the goal is to teach it concepts about music theory, while still maintaining the information about typical melodies originally learned from data. To accomplish this task, we propose *RL Tuner*, a novel sequence training method incorporating RL. We use an LSTM trained on data (the *Note RNN*) to supply the initial weights for three networks in *RL Tuner*: the *Q-network* and *Target Q-network* in the DQN algorithm as described in Section 2.1, and a *Reward RNN*. Therefore, the *Q-network* is a recurrent LSTM model, with architecture identical to that of the original *Note RNN*. The *Reward RNN* is used to supply part of the reward value used to train the model, and is held fixed during training.

In order to formulate music generation as an RL problem, we treat placing the next note in the melody as taking an action. The state of the environment $s$ consists of the previous note, and the internal state of the LSTM cells of both the *Q-network* and the *Reward RNN*. Thus, $Q(a, s)$ can be calculated by initializing the recurrent *Q-network* with the appropriate memory cell contents, running it for one time step using the previous note, and evaluating the output value for the action $a$. The next action can be selected with either a Boltzmann sampling or $\epsilon$-greedy exploration strategy.

Given action $a$, the reward can be computed by combining probabilities learned from the training data with knowledge of music theory. We define a set of music-theory based rules (described in Section 3.2) to impose constraints on the melody that the model is composing through a reward signal $r_{MT}(a, s)$. For example, if a note is in the wrong key, then the model receives a negative reward. However, it is necessary that the model still be "creative," rather than learning a simple melody that can easily exploit these rewards. Therefore, we use the *Reward RNN* — or equivalently the trained *Note RNN* — to compute $\log p(a|s)$, the log probability of a note $a$ given a melody $s$, and incorporate this into the reward function. Figure 1 illustrates these ideas.

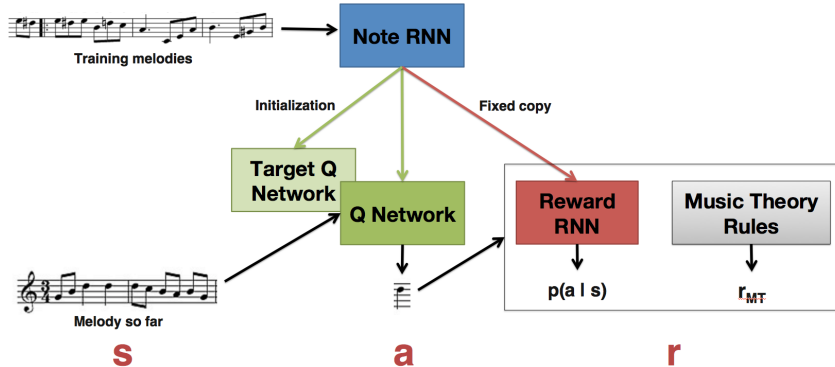

Figure 1: A *Note RNN* is trained on MIDI files and supplies the initial weights for the *Q-network* and *Target-Q-network*, and final weights for the *Reward RNN*.

The total reward given at time $t$ is therefore:

$$r(s, a) = \log p(a|s) + r_{MT}(a, s)/c \tag{3}$$

where $c$ is a constant controlling the emphasis placed on the music theory reward. Given the DQN loss function in Eq. 2 and modified reward function in Eq. 3, the new loss function and learned policy for *RL Tuner* are,

$$L(\theta) = \mathbb{E}_\beta[(\log p(a|s) + r_{MT}(a, s)/c + \gamma \max_{a'} Q(s', a'; \theta^-) - Q(s, a; \theta))^2] \tag{4}$$

$$\pi_\theta(a|s) = \delta(a = \arg\max_a Q(s, a; \theta)). \tag{5}$$

Thus, the modified loss function forces the model to learn that the most valuable actions are those that conform to the music theory rules, but still have high probability in the original data.

### 3.1    RELATIONSHIP TO KL CONTROL

The technique described in Section 3 has a close connection to stochastic optimal control (SOC) (Stengel, 1986) and in particular, KL control (Todorov, 2006; Kappen et al., 2012; Rawlik et al., 2012). SOC casts the optimal planning in stochastic environments as inference in graphical models, and enables direct application of probabilistic inference techniques such as Expectation-Maximization (EM) and message passing for solving the control problem (Attias, 2003; Toussaint & Storkey, 2006; Toussaint, 2009). Rawlik et al. (2012); Kappen et al. (2012) then introduced KL control, a generic formulation of the SOC as Kullback-Leibler (KL) divergence minimization, and connected to prior work on RL with additional KL cost (Todorov, 2006). Since our primary focus is to connect with DQNs, we specifically focus on the work by Rawlik et al. (2012) as they derive a temporal-difference-based approach on which we build our methods.

KL control formulation defines a prior dynamics or policy, and derives a variant of the control or RL problem as performing approximate inference in a graphical model. Let $\tau$ be a trajectory of state and action sequences, $p(\tau)$ be a prior dynamics, and $r(\tau)$ be the reward of the trajectory. Then, an additional binary variable $b$ is introduced and a graphical model is defined as $p(\tau, b) = p(\tau)p(b|\tau)$, where $p(b = 1|\tau) = e^{r(\tau)/c}$ and $c$ is the temperature variable. An approximation to $p(\tau|b = 1)$ can be derived using the variational free-energy method, and this leads to a cost with a similar form to the RL problem previously defined, but with an additional penalty based on the KL divergence from the prior trajectory,

$$\log p(\tau|b = 1) = \log \int p(\tau)p(b|\tau)d\tau \tag{6}$$

$$\geq \mathbb{E}_{q(\tau)}[\log p(\tau)p(b|\tau) - \log q(\tau)] \tag{7}$$

$$= \mathbb{E}_{q(\tau)}[r(\tau)/c - \mathbf{KL}[q(\tau)||p(\tau)]] = L_v(q) \tag{8}$$

where $q(\tau)$ is the variational distribution. Rewriting the variational objective $L_v(q)$ in Eq. 6 in terms of policy $\pi_\theta$, we get the following RL objective with KL-regularization, also known as KL control,

$$L_v(\theta) = \mathbb{E}_\pi[\sum_t r(s_t, a_t)/c - KL[\pi_\theta(\cdot|s_t)||p(\cdot|s_t)]]. \tag{9}$$

In contrast, the objective in Section 3 is,

$$L_v(\theta) = \mathbb{E}_\pi[\sum_t r(s_t, a_t)/c + \log p(a_t|s_t)]. \tag{10}$$

The difference is that Eq. 9 includes an entropy regularizer, and thus a different off-policy method from $Q$-learning is required. A generalization of $\Psi$-learning (Rawlik et al., 2012), and $G$-learning (Fox et al.)[2] are two off-policy methods for solving the KL-regularized RL problem, where additional generalized-$\Psi$ and $G$ functions are defined and learned instead of $Q$. We implement both of these algorithms as well, treating the prior policy as the conditional distribution $p(a|s)$ defined by the trained *Note RNN*. To the best of our knowledge, this is the first application of KL-regularized off-policy methods with deep neural networks to sequence modeling tasks. The two methods are given below respectively,

$$L(\theta) = \mathbb{E}_\beta[(\log p(a|s) + r_{MT}(s, a)/c + \gamma \log \sum_{a'} e^{\Psi(s', a'; \theta^-)} - \Psi(s, a; \theta))^2] \tag{11}$$

$$\pi_\theta(a|s) \propto e^{\Psi(s, a; \theta)} \tag{12}$$

$$L(\theta) = \mathbb{E}_\beta[(r_{MT}/c(s, a) + \gamma \log \sum_{a'} e^{\log p(a'|s') + G(s', a'; \theta^-)} - G(s, a; \theta))^2] \tag{13}$$

$$\pi_\theta(a|s) \propto p(a|s)e^{G(s, a; \theta)}. \tag{14}$$

---

[2]The methods in the original papers are derived for different motivations and presented in different forms as described in Section 4, but we refer them using their names as the derivations follow closely from the papers.

Both methods can be seen as instances of KL-regularized deep Q-learning, and they also subsume entropy-regularized deep Q-learning by removing the $\log p(a|s)$ term. The main difference between the two methods is the definition of the action-value functions generalized-$\Psi$ and $G$. In fact $G$-learning can be directly derived from generalized $\Psi$-learning by reparametrizing $\Psi(s,a) = \log p(a|s) + G(s,a)$. The $G$-function does not give the policy directly but instead needs to be dynamically mixed with the prior policy probabilities. While this computation is straight-forward for discrete action domains as here, extensions to continuous action domains require additional considerations such as normalizability of advantage function parametrizations (Gu et al., 2016). The KL control-based derivation also has another benefit in that the stochastic policies can be directly used as an exploration strategy, instead of heuristics such as $\epsilon$-greedy or additive noise (Mnih et al., 2013; Lillicrap et al., 2016). The derivations for both methods are included in the appendix for completeness.

### 3.2 Music-theory based reward

A central question of this paper is whether RL can be used to constrain a sequence learner such that the sequences it generates adhere to a desired structure. To test this hypothesis, we developed several rules that we believe describe more pleasant-sounding melodies, taking inspiration from a text on melodic composition (Gauldin, 1995). We do not claim these characteristics are exhaustive, strictly necessary for good composition, or even particularly interesting. They simply serve the purpose of guiding the model towards traditional composition structure. It is therefore crucial to apply the *RL Tuner* framework to retain the knowledge learned from real songs in the training data.

Following the principles set out on page 42 of Gauldin's book (Gauldin, 1995), we define the reward function $r_{MT}(a,s)$ to encourage melodies to have the following characteristics. All notes should belong to the same key, and the melody should begin and end with the tonic note of the key; e.g. if the key is C-major, this note would be middle C. This note should occur in the first beat and last 4 beats of the melody. Unless a rest is introduced or a note is held, a single tone should not be repeated more than four[3] times in a row. To encourage variety, we penalize the model if the melody is highly correlated with itself at a lag of 1, 2, or 3 beats. The penalty is applied when the auto-correlation coefficient is greater than .15. The melody should avoid awkward intervals like augmented 7ths, or large jumps of more than an octave. Gauldin also indicates good compositions should move by a mixture of small steps and larger harmonic intervals, with emphasis on the former; the reward values for intervals reflect these requirements. When the melody moves with a large interval (a 5th or more) in one direction, it should eventually be resolved by a leap back or gradual movement in the opposite direction. Leaping twice in the same direction is negatively rewarded. The highest note of the melody should be unique, as should the lowest note. Finally, the model is rewarded for playing *motifs*, which are defined as a succession of notes representing a short musical "idea"; in our implementation, a bar of music with three or more unique notes. Since repetition has been shown to be key to emotional engagement with music (Livingstone et al., 2012), we also sought to train the model to repeat the same motif within a melody.

## 4 Related Work

Generative modeling of music with RNNs has been explored in a variety of contexts, including generating Celtic folk music (Sturm et al., 2016), or performing Blues improvisation (Eck & Schmidhuber, 2002). Other approaches have examined RNNs with richer expressivity, latent-variables for notes, or raw audio synthesis (Boulanger-Lewandowski et al., 2012; Gu et al., 2015; Chung et al., 2015). Recently, impressive performance in generating music from raw audio has been attained with convolutional neural networks with receptive fields at various time scales (Dieleman et al., 2016).

Although the application of RL to RNNs is a relatively new area, recent work has attempted to combine the two approaches. MIXER (Mixed Incremental Cross-Entropy Reinforce) (Ranzato et al., 2015) uses BLEU score as a reward signal to gradually introduce a RL loss to a text translation model. After initially training the model using cross-entropy, the training process is repeated using cross-entropy loss for the $T - \Delta$ tokens in a sequence (where $T$ is the length of the sequence), and

---

[3]While the number four can be considered a rough heuristic, avoiding excessively repeated notes and static melodic contours is Gauldin's first rule of melodic composition (Gauldin, 1995).

using RL for the remainder of the sequence. Another approach (Bahdanau et al., 2016) applies an actor-critic method and uses BLEU score directly to train a critic network to output the value of each word, where the actor is again initialized with the policy of an RNN trained with next-step prediction. Reward-augmented maximum likelihood (Norouzi et al., 2016) augments the standard ML with a sequence-level reward function and connects it with the above RL training methods. These approaches assume that the complete task reward specification is available. They pre-train a good policy with supervised learning so that RL can be used to learn with the true task objective, since training with RL from scratch is difficult. *RL Tuner* instead only uses rewards to correct certain properties of the generated data, while learning most information from data. This is important since in many sequence modeling applications such as music or language generation, the true reward function is not available or imperfect and ultimately the model should rely on learning from data. The *RL Tuner* method provides an elegant and flexible framework for correcting undesirable behaviors of RNNs that can arise from limited training data or imperfect training algorithms.

SeqGAN (Yu et al., 2016) applies RL to an RNN by using a discriminator network — similar to those used in Generative Adversarial Networks (GANs) (Goodfellow et al., 2014) — to classify the realism of a complete sequence, and this classifier-based reward is used as a reward signal to the RNN. The approach is applied to a number of generation problems, including music generation. Although the model obtained improved MSE and BLEU scores on the Nottingham music dataset, it is not clear how these scores map to the subjective quality of the samples (Huszár, 2015), and no samples are provided with the paper. In contrast, we provide both samples and quantitative results demonstrating that our approach improves the metrics defined by the reward function. Further, we show that *RL Tuner* can be used to explicitly correct undesirable behaviors of an RNN, which could be useful in a broad range of applications.

Also related to our work is that of Li and colleagues Li et al. (2016), in which the authors pre-train a model with MLE and then use RL to impose heuristic rules designed to improve the dialog generated by the model. However, after pre-training, only the heuristic rewards are used for further training, which alters the model to optimize only for the heuristic rewards, whereas our approach allows the model to retain information learned from data, while explicitly controlling the trade-off between the influence of data and heuristic reward with the $c$ parameter. While Li and colleagues do use the outputs of the pre-trained model as part of one of the heuristic reward functions, it is only to teach the model to choose dialog turns that minimize the probability that the pre-trained model places on "dull" responses, such as "I don't know". However, our approach directly penalizes divergence from the probability distribution learned by the MLE model for every response, allowing the model to retain information about the full space of sequences originally learned from data.

Finally, as discussed in Section 3.1, our approach is related to stochastic optimal control (SOC) (Stengel, 1986) and KL control (Todorov, 2006; Kappen et al., 2012; Rawlik et al., 2012), in particular the two off-policy, model-free methods, $\Psi$-learning (Rawlik et al., 2012) and $G$-learning (Fox et al.). Both approaches solve a KL-regularized RL problem, in which a term is introduced to the reward objective to penalize KL divergence from some prior policy. While our methods rely on similar derivations presented in these papers, there are some key differences. First, these techniques have not been applied to DQNs or RNNs, or as a way to fine-tune a pre-trained RNN with additional desired charateristics. Secondly, our methods have different motivations and forms from the original papers: original $\Psi$-learning (Rawlik et al., 2012) restricts the prior policy to be the policy at the previous iteration and solves the original RL objective with conservative, KL-regularized policy updates, similar to conservative policy gradient methods (Kakade, 2001; Peters et al., 2010; Schulman et al., 2015). The original $G$-learning (Fox et al.) penalizes divergence from a simple uniform prior policy in order to cope with over-estimation of target $Q$ values, and includes scheduling for the temperature parameter $c$. Lastly, our work includes the $Q$-learning objective with additional cross-entropy reward as a comparable alternative, and provides for the first time comparisons among the three methods for incorporating prior knowledge in RL.

## 5 EXPERIMENTS

To train the *Note RNN*, we extract monophonic melodies from a corpus of 30,000 MIDI songs. Melodies are quantized at the granularity of a sixteenth note, so each time step corresponds to one sixteenth of a bar of music. We encode a melody using two special events plus three octaves of notes.

The special events are used to introduce rests and notes with longer durations, and are encoded as $0 = $ *note off*, $1 = $ *no event*. Three octaves of pitches, starting from MIDI pitch 48, are then encoded as $2 = $ C3, $3 = $ C#3, $4 = $ D3, ..., $37 = $ B5. For example, the sequence $\{4, 1, 0, 1\}$ encodes an eighth note with pitch D3, followed by an eighth note rest. As the melodies are monophonic, playing another note implicitly ends the last note that was played without requiring an explicit *note off* event. Thus the sequence $\{2, 4, 6, 7\}$ encodes a melody of four sixteenth notes: C3, D3, E3, F3. A length-38 one-hot encoding of these values is used for both network input and network output.

The *Note RNN* consists of one LSTM layer of 100 cells, and was trained for 30,000 iterations with a batch size of 128. Optimization was performed with Adam (Kingma & Ba, 2014), and gradients were clipped to ensure the L2 norm was less than 5. The learning rate was initially set to .5, and a momentum of 0.85 was used to exponentially decay the learning rate every 1000 steps. To regularize the network, a penalty of $\beta = 2.5 \times 10^{-5}$ was applied to the L2 norm of the network weights. Finally, the losses for the first 8 notes of each sequence were not used to train the model, since it cannot reasonably be expected to accurately predict them with no context. The trained *Note RNN* eventually obtained a validation accuracy of 92% and a log perplexity score of .2536.

The learned weights of the *Note RNN* were used to initialize the three sub-networks in the *RL Tuner* model. Each *RL Tuner* model was trained for 1,000,000 iterations, using the Adam optimizer, a batch size of 32, and clipping gradients in the same way. The reward discount factor was $\gamma$=.5. The Target-$Q$-network's weights $\theta^-$ were gradually updated to be similar to those of the $Q$-network ($\theta$) according to the formula $(1 - \eta)\theta^- + \eta\theta$, where $\eta = .01$ is the Target-$Q$-network update rate. We replicated our results for a number of settings for the weight placed on the music-theory rewards, $c$; we present results for $c$=.5 below because we believe them to be most musically pleasing. Similarly, we replicated the results using both $\epsilon$-greedy and Boltzmann exploration, and present the results using $\epsilon$-greedy exploration below.

We compare three methods for implementing *RL Tuner*: $Q$-learning, generalized $\Psi$-learning, and $G$-learning, where the policy defined by the trained *Note RNN* is used as the cross entropy reward in $Q$-learning and the prior policy in $G$- and generalized $\Psi$-learning. These approaches are compared to both the original performance of the *Note RNN*, and a model trained using only RL and no prior policy. Model evaluation is performed every 100,000 training epochs, by generating 100 melodies and assessing the average $r_{MT}$ and $\log p(a|s)$.

All of the code for *RL Tuner*, including a checkpointed version of the trained *Note RNN* is available at https://github.com/natashamjaques/magenta/tree/rl-tuner.

## 6    RESULTS

Table 1 provides quantitative results in the form of performance on the music theory rules to which we trained the model to adhere; for example, we can assess the fraction of notes played by the model which belonged to the correct key, or the fraction of melodic leaps that were resolved. The statistics were computed by randomly generating 100,000 melodies from each model.

| Metric | Note RNN | Q | $\Psi$ | G |
|---|---|---|---|---|
| Notes excessively repeated | 63.3% | **0.0%** | **0.02%** | **0.03%** |
| Mean autocorrelation - lag 1 | -.16 | **-.11** | **-.10** | .55 |
| Mean autocorrelation - lag 2 | .14 | **.03** | **-.01** | .31 |
| Mean autocorrelation - lag 3 | -.13 | **.03** | **.01** | 17 |
| Notes not in key | 0.1% | 1.00% | 0.60% | 28.7% |
| Melodies starting with tonic | 0.9% | **28.8%** | **28.7%** | 0.0% |
| Leaps resolved | 77.2% | **91.1%** | **90.0%** | 52.2% |
| Melodies with unique max note | 64.7% | 56.4% | 59.4% | 37.1% |
| Melodies with unique min note | 49.4% | 51.9% | **58.3%** | **56.5%** |
| Notes in motif | 5.9% | **75.7%** | **73.8%** | **69.3%** |
| Notes in repeated motif | 0.007% | **0.11%** | **0.09%** | 0.01% |

Table 1: Statistics of music theory rule adherence based on 100,000 randomly initialized melodies generated by each model. The top half of the table contains metrics that should be near zero, while the bottom half contains metrics that should increase. Bolded entries represent significant improvements over the *Note RNN* baseline.

The results above demonstrate that the application of RL is able to correct almost all of the targeted "bad behaviors" of the *Note RNN*, while improving performance on the desired metrics. For example, the original LSTM model was extremely prone to repeating the same note; after applying RL, we see that the number of notes belonging to some excessively repeated segment has dropped from 63% to nearly 0% in all of the *RL Tuner* models. While the metrics for the G model did not improve as consistently, the $Q$ and $\Psi$ models successfully learned to play in key, resolve melodic leaps, and play motifs. The number of melodies that start with the tonic note has also increased, melody auto-correlation has decreased, and repeated motifs have increased slightly. The degree of improvement on these metrics is related to the magnitude of the reward given for the behavior. For example, a strong penalty of -100 was applied each time a note was excessively repeated, while a reward of only 3 was applied at the end of a melody for unique extrema notes (which most likely explains the lack of improvement on this metric). The reward values could be adjusted to improve the metrics further, however we found that these values produced the most pleasant melodies.

While the metrics indicate that the targeted behaviors of the RNN have improved, it is not clear whether the models have retained information about the training data. Figure 2a plots the average $\log p(a|s)$ as produced by the *Reward RNN* for melodies generated by the models every 100,000 training epochs; Figure 2b plots the average $r_{MT}$. Included in the plots is an *RL only* model trained using only the music theory rewards, with no information about $\log p(a|s)$. Since each model is initialized with the weights of the trained *Note RNN*, we see that as the models quickly learn to adhere to the music theory constraints, $\log p(a|s)$ falls from its initial point. For the *RL only* model, $\log p(a|s)$ reaches an average of -3.65, which is equivalent to an average $p(a|s)$ of approximately 0.026. Since there are 38 actions, this represents essentially a random policy with respect to the distribution defined by the *Note RNN*. Figure 2a shows that each of our models ($Q$, $\Psi$, and $G$) attain higher $\log p(a|s)$ values than this baseline, indicating they have maintained information about the data probabilities. The $G$-learning implementation scores highest on this metric, at the cost of slightly lower average $r_{MT}$. This compromise between data probability and adherence to music theory could explain the difference in $G$ model's performance on the music theory metrics in Table 1. Finally, while $c = 0.5$ produced melodies that sounded better subjectively, we found that by increasing the $c$ parameter it is possible to train all the models to have even higher average $\log p(a|s)$.

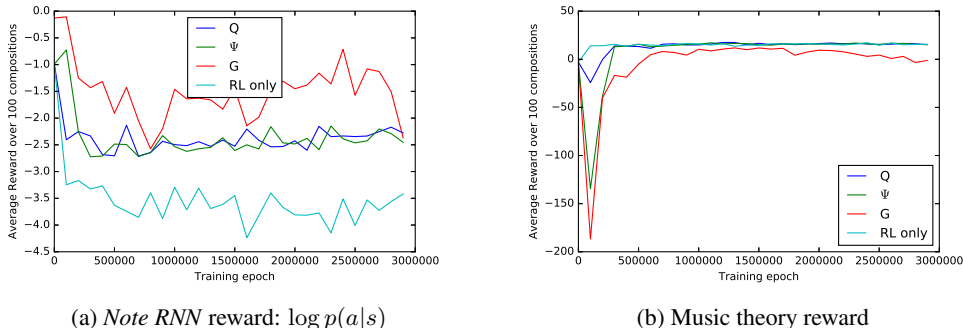

(a) *Note RNN* reward: $\log p(a|s)$          (b) Music theory reward

Figure 2: Average reward obtained by sampling 100 melodies every 100,000 training epochs. The three models are compared to a model trained using only the music theory rewards $r_{MT}$.

The question remains whether the RL-tuned models actually produce more pleasing melodies. To answer it, we conducted a user study via Amazon Mechanical Turk in which participants were asked to rate which of two randomly selected melodies they preferred on a Likert scale. A total of 192 ratings were collected; each model was involved in 92 of these comparisons. Figure 3 plots the number of comparisons in which a melody from each model was selected as the most musically pleasing. A Kruskal-Wallis H test of the ratings showed that there was a statistically significant difference between the models, $\chi^2(3) = 109.480, p < 0.001$. Mann-Whitney U post-hoc tests revealed that the melodies from all three *RL Tuner* models ($Q$, $\Psi$, and $G$) had significantly higher ratings than the melodies of the *Note RNN*, $p < .001$. The $Q$ and $\Psi$ melodies were also rated as significantly more pleasing than those of the $G$ model, but did not differ significantly from each other. The sample melodies used for the study are available here: goo.gl/XIYt9m; we encourage readers to judge their quality for themselves.

Listening to the samples produced by the *Note RNN* reveals that they are sometimes dischordant and usually dull; the model tends to place rests frequently, repeat the same note, and produce melodies with little variation. In contrast, the melodies produced by the *RL Tuner* models are more varied and interesting. The $G$ model tends to produce energetic and chaotic melodies, which include sequences of repeated notes. This repetition is likely because the G policy as defined in Eq. 14 directly mixes $p(a|s)$ with the output of the G network, and the *Note RNN* strongly favours repeating notes. The most pleasant-sounding melodies are generated by the $Q$ and $\Psi$ models. These melodies stay firmly in key and frequently choose more harmonious interval steps, leading to melodic and pleasant melodies. However, it is clear they have retained information about the training data; for example, the sample q2.wav in the sample directory ends with a seemingly familiar riff.

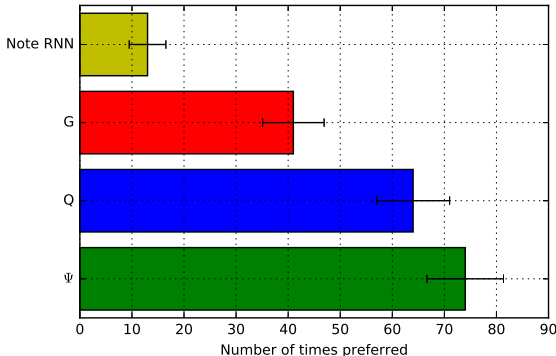

Figure 3: The number of times a melody from each model was selected as most musically pleasing. Error bars reflect the std. dev. of a binomial distribution fit to the binary win/loss data from each model.

# 7 DISCUSSION AND FUTURE WORK

We have derived a novel sequence learning framework which uses RL rewards to correct properties of sequences generated by an RNN, while keeping much of the information learned from supervised training on data. We proposed and evaluated three alternative techniques for achieving this, and showed promising results on music generation tasks.

While we acknowledge that the simple monophonic melodies generated by these models — which are based on overly simplistic rules of melodic composition — do not approach the level of artistic merit of human composers, we believe this study provides a proof-of-concept that encoding domain knowledge using our method can help the outputs of an LSTM adhere to a more consistent structure. The musical complexity of the songs is limited not just by the heuristic rules, but also by the numerical encoding, which cannot represent the dynamics and expressivity of a musical performance. However, although these simple melodies cannot surpass those of human musicians, attempting to train a model to generate aesthetically pleasing outputs in the absence of a better metric of human taste than log-likelihood is a problem of broader interest to the artificial intelligence community.

In addition to the ability to train models to generate pleasant-sounding melodies, we believe our approach of using RL to refine RNN models could be promising for a number of applications. For example, it is well known that a common failure mode of RNNs is to repeatedly generate the same token. In text generation and automatic question answering, this can take the form of repeatedly generating the same response (e.g. "How are you?" → "How are you?" → "How are you?" ...). We have demonstrated that with our approach we can correct for this unwanted behavior, while still maintaining information that the model learned from data. Although manually writing a reward function may seem unappealing to those who believe in training models end-to-end based only on data, that approach it is limited by the quality of the data that can be collected. If the data contains hidden biases, this can lead to highly undesirable consequences. Recent research has shown that the *word2vec* embeddings in popular language models trained on standard corpora consistently contain the same harmful biases with respect to race and gender that are revealed by implicit association tests on humans (Caliskan-Islam et al., 2016). In contrast to relying solely on possibly biased data, our approach allows for encoding high-level domain knowledge into the RNN, providing a general, alternative tool for training sequence models.

ACKNOWLEDGMENTS

This work was supported by Google Brain, the MIT Media Lab Consortium, and Canada's Natural Sciences and Engineering Research Council (NSERC). We thank Dzmitry Bahdanau, Greg Wayne, Sergey Levine, and Timothy Lillicrap for helpful discussions on RL and stochastic optimal control.

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

# 8 APPENDIX

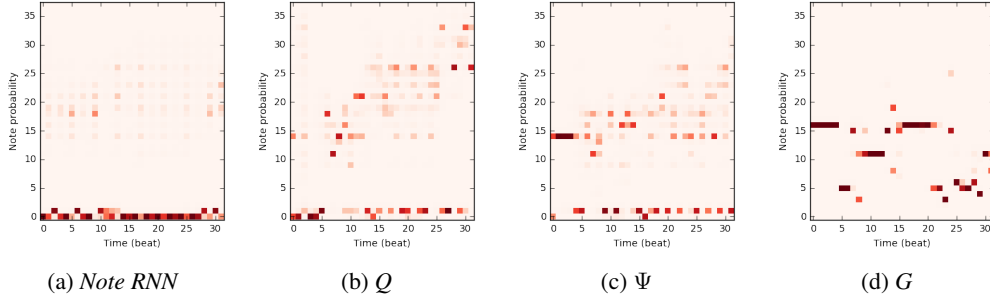

(a) *Note RNN*       (b) $Q$       (c) $\Psi$       (d) $G$

Figure 4: Probability distribution over the next note generated by each model for a sample melody. Probability is shown on the vertical axis, with red indicating higher probability. Note 0 is *note off* and note 1 is *no event*.

## 8.1 OFF-POLICY METHODS DERIVATIONS FOR KL-REGULARIZED REINFORCEMENT LEARNING

Given the KL-regularized RL objective defined in Eq. 9, the value function is given by,

$$V(s_t; \pi) = \mathbb{E}_\pi[\sum_{t' \geq t} r(s_{t'}, a_{t'})/c - \text{KL}[\pi(\cdot|s_{t'})||p(\cdot|s_{t'})]] \tag{15}$$

### 8.1.1 GENERALIZED $\Psi$-LEARNING

The following derivation is based on modifications to (Rawlik et al., 2012) and resembles the derivation in Fox et al.. We define the generalized $\Psi$ function as,

$$\Psi(s_t, a_t; \pi) = r(s_t, a_t)/c + \log p(a_t|s_t) \tag{16}$$

$$+ \mathbb{E}_{p(s_{t+1}|s_t,a_t)}\mathbb{E}_\pi[\sum_{t' \geq t+1} r(s_{t'}, a_{t'})/c - \text{KL}[\pi(\cdot|s_{t'})||p(\cdot|s_{t'})]] \tag{17}$$

$$= r(s_t, a_t)/c + \log p(a_t|s_t) + \mathbb{E}_{p(s_{t+1}|s_t,a_t)}[V(s_{t+1}; \pi)] \tag{18}$$

The value function can be expressed as,

$$V(s_t; \pi) = \mathbb{E}_\pi[\Psi(s_t, a_t; \pi)] + \mathbb{H}[\pi] \tag{19}$$

$$= \mathbb{E}_\pi[\Psi(s_t, a_t; \pi) - \log \pi(a_t|s_t)] \tag{20}$$

Fixing $\Psi(s_t, a_t) = \Psi(s_t, a_t; \pi)$ and constraining $\pi$ to be a probability distribution, the optimal greedy policy update $\pi^*$ can be derived by functional calculus, along with the corresponding optimal value function,

$$\pi^*(a_t|s_t) \propto e^{\Psi(s_t,a_t)} \tag{21}$$

$$V(s_t; \pi^*) = \log \sum_{a_t} e^{\Psi(s_t,a_t)} \tag{22}$$

Given Eq. 18 and 22, the following Bellman optimality equation for generalized $\Psi$ function is derived, and the $\Psi$-learning loss in Eq. 11 directly follows.

$$\Psi(s_t, a_t; \pi^*) = r(s_t, a_t)/c + \log p(a_t|s_t) + \mathbb{E}_{p(s_{t+1}|s_t,a_t)}[\log \sum_{a_{t+1}} e^{\Psi(s_{t+1},a_{t+1};\pi^*)}] \tag{23}$$

### 8.1.2 $G$-LEARNING

The following derivation is based on (Fox et al.) with small modifications. We define the $G$ function as,

$$G(s_t, a_t; \pi) = r(s_t, a_t)/c + \mathbb{E}_{p(s_{t+1}|s_t,a_t)}\mathbb{E}_\pi[\sum_{t' \geq t+1} r(s_{t'}, a_{t'})/c - \text{KL}[\pi(\cdot|s_{t'})||p(\cdot|s_{t'})]] \tag{24}$$

$$= r(s_t, a_t)/c + \mathbb{E}_{p(s_{t+1}|s_t,a_t)}[V(s_{t+1}; \pi)] = \Psi(s_t, a_t; \pi) - \log p(a_t|s_t) \tag{25}$$

Similar derivation as above can be applied.

$$V(s_t; \pi) = \mathbb{E}_\pi[G(s_t, a_t; \pi)] - \text{KL}[\pi(\cdot|s_{t'})||p(\cdot|s_{t'})] \tag{26}$$

$$= \mathbb{E}_\pi[G(s_t, a_t; \pi) - \frac{\log \pi(a_t|s_t)}{\log p(a_t|s_t)}] \tag{27}$$

$$\pi^*(a_t|s_t) \propto p(a_t|s_t)e^{G(s_t, a_t)} \tag{28}$$

$$V(s_t; \pi^*) = \log \sum_{a_t} p(a_t|s_t)e^{G(s_t, a_t)} \tag{29}$$

$$G(s_t, a_t; \pi^*) = r(s_t, a_t)/c + \mathbb{E}_{p(s_{t+1}|s_t, a_t)}[\log \sum_{a_{t+1}} p(a_{t+1}|s_{t+1})e^{G(s_{t+1}, a_{t+1}; \pi^*)}] \tag{30}$$

Alternatively, the above expression for $G$-learning can be derived from $\Psi$-learning by simple reparametrization with $\Psi(s, a) = G(s, a) + \log p(a|s)$ in Eq. 23.

