# Peer review of "Tuning Recurrent Neural Networks with Reinforcement Learning"

_ICLR 2017 — rejected_

[Public Comment · Aravind Lakshminarayanan · 06 Nov 2016]
**Nice work! Similar work exists for Dialog generation**

Nice work! There has been work on tuning RNNs that have been trained with supervised learning with additional "engineered" reward functions and using RL (policy gradients). Check out

[Public Comment · Roy Fox · 02 Dec 2016]
**Very nice work! A few comments:**

Very nice work!
Interesting to see how well it works to combine a reward signal with cross-entropy from an informative prior, even without the extra entropy regularization of the KL cost.

We would like to note a few important things, though:

1. Psi-learning is not the method given in (11), (12), and Appendix 8.1.1.
Using the current policy as the prior for each small policy update is central to Psi-learning and for the properties of the Psi function.
This is best seen by comparing your (23) with Rawlik et al.'s (14), where they have Psi-\bar{Psi} substituted for log(p).
By having p fixed and not dependent of Psi, you are qualitatively changing the fixed point of the equation.
For example, their Psi function is gap increasing, in that it only recovers the value function in the optimal action, and is negative infinity for suboptimal actions — your "Psi" function does not have this property, and the two should not be confused.

2. Psi-learning is distinct from G-learning, in that the former uses small divergences from a changing prior, while the latter uses large divergences from a fixed prior.
By fixing the prior p in both of your derivations, both are using G-learning with a fixed temperature, albeit with different parameterization of the function G.
The two functions defined in (16) and (24) differ by the constant log(p), and this is then reflected in (23) and (30).
Intriguingly, parameterizing G+log(p) yields better results than parameterizing G.
This is somewhat reminiscent of advantage learning, where one parameterizes Q-V rather than Q, and certainly merits further study.

3. In DQN, the Q-network is a feed-forward network that maps (s,a) inputs into real outputs.
If this is the architecture used here, it is unclear how it is initialized from the differently shaped Note RNN.
On other hand, if the Q-network here is a RNN, it is unclear what state it keeps and how it computes Q(s,a;theta).

4. As Figure 2 shows, "G-learning" achieves a different tradeoff with worse music-theory reward but better log(p).
It is therefore inaccurate to characterize it in Section 6 and Table 1 as doing worse.
For a fair comparison, one needs to set c differently for each algorithm, so that one reward is the same, and compare the other reward.
It would also be useful to repeat this for various values of c, and plot the two rewards on a plane, to allow comparing the reward-achievability regions of the algorithms.

5. Stochastic optimal control is a much broader field (cf. Bertsekas 1995). In Section 3.1 you are referring specifically to the field known as KL control.


Ari Pakman and Roy Fox

[Official Review · AnonReviewer4 · rating 5 · confidence 5 · 16 Dec 2016 (modified: 21 Jan 2017)]
**Log-likelihood models augmented by non-differentiable objectives for MIDI synthesis**

The authors propose a solution for the task of synthesizing melodies. The authors claim that the "language-model"-type approaches with LSTMs generate melodies with certain shortcomings. They tend to lack long-range structure, to repeat notes etc. To solve this problem the authors suggest that the model could be first trained as a pure LM-style LSTM and then trained with reinforcement learning to optimize an objective which includes some non-differentiable music-theory related constraints. 

The reinforcement learning methodology is appropriate but straightforward and closely resembles previous work for text modeling and dialogue generation. By itself the methodology doesn't offer a new technique. 

To me, the paper's contribution then comes down to the novelty / utility / impact of the application. The authors clearly put substantial of effort into crafting the rules and user study and that is commendable. On the other hand, music itself is dealt with somewhat naively. While the user study reflects hard work, it seems premature. The semi-plausible piano melodies here are only music in the way that LSTM Shakespeare passes as poetry. So it's analogous to conducting a user study comparing LSTM Shakespeare to n-gram Shakespeare. 

I'd caution the author's against the uncritical motivation that a problem has previously been studied. Research contains abundant dead ends (not to say this is necessarily one) and the burden to motivate research shouldn't be forgotten. This is especially true when the application is the primary thrust of a paper.

Generally the authors should be careful about describing this model as "composing". By analogy to a Shakespeare-LSTM, the language model is not really composing English prose. The relationship between constructing a statistical sequence model and creating art - an activity that involves communication grounded in real-world semantics should not be overstated. 

I appreciate the authors' efforts to respond to some criticisms of the problem setup and encourage them to anticipate these arguments in the paper and to better motivate the work in the future. If the main contribution is the application (the methods have been used elsewhere), then the motivation is of central importance. I also appreciate their contention that the field benefits from multiple datasets and not simply relying on language modeling. Further, they are correct in asserting that MIDI can capture all the information in a score (not merely "Gameboy music", and that for some musics (e.g. European classical) the score is of central importance. However, the authors may overstate the role of a score in jazz music.

Overall, for me, the application, while fun, doesn't add enough to the impact of the paper. And the methodology, while appropriate, doesn't stand on its own. 

--Update-- Thanks for your modifications and arguments. I've revised my scores to add a point.

[Official Review · AnonReviewer1 · rating 5 · confidence 5 · 16 Dec 2016]
**Combining data driven models and reinforcement signals**

This paper uses a combination of likelihood and reward based learning to learn sequence models for music. The ability to combine likelihood and reward based learning has been long known, as a result of the unification of inference and learning first appearing in the ML literature with the EM formalism of Attias (2003) for fixed horizons, extended by Toussaint and Storkey (2006), to general horizon settings, Toussaint et al. (2011) to POMDPs and generalised further by Kappen et Al. (2012) and Rawlik et Al. (2012). These papers introduced the basic unification, and so any additional probabilistic or data driven objective can be combined with the reinforcement learning signal: it is all part of a unified reward/likelihood. Hence the optimal control target under unification is p(b=1|\tau)E_p(A,S) \prod_t \pi(a_t|s_t): i.e. the probability of getting reward, and probability of the policy actions under the known data-derived distribution, thereby introducing the log p(a_t|s_t) into (9) too.

The interpretation of the secondary objective as the prior is an alternative approach under a stochastic optimal control setting, but not the most natural one given the whole principle of SOC of matching control objectives to inference objectives. The SOC off policy objective still does still contain the KL term so the approach would still differ from the approach of this paper.

Though the discussion of optimal control is good, I think some further elaboration of the history and how reward augmentation can work in SOC would be valuable. This would allow SOC off-policy methods to be compared with the DQN directly, like for like.

The motivation of the objective (3) is sensible but could be made clearer via the unification argument above. Then the paper uses DCN to take a different approach from the variational SOC for achieving that objective.

Another interesting point of discussion is the choice of E_pi \log p(a_t|s_t) – this means the policy must “cover” the model. But one problem in generation is that a well-trained model is often underfit, resulting in actions that, over the course of a number of iterations, move the state into data-unsupported parts of the space. As a result the model is no longer confident and quickly tends to be fairly random. This approach (as opposed to a KL(p||pi) – which is not obvious how to implement) cannot mitigate against that, without a very strong signal (to overcome the tails of a distribution). In music, with a smaller discrete alphabet, this is likely to be less of a problem than for real valued policy densities, with exponentially decaying tails. Some further discussion of what you see in light of this issue would be valuable: the use of c to balance things seems critical, and it seems clear from Figure 2 that the reward signal needed to be very high to push the log p signal into the right range.

Altogether, in the music setting this paper provides a reasonable demonstration that augmentation of a sequence model with an additional reward constraint is valuable. It demonstrates that DQN is one way of learning that signal, but AFAICS it does not compare learning the same signal via other techniques. Instead for the comparator techniques it reverts to treating the p(a|s) as a “prior” term rather than a reward term, leaving a bit of a question as to whether DQN is particularly appropriate. 

Another interesting question for the discussion is whether the music theory reward could be approximated by a differentiable model, mitigating the need for an RL approach at all.

[Official Review · AnonReviewer2 · rating 6 · confidence 3 · 19 Dec 2016]
**Hand-crafted musical reward for fine-tuning LSTMs**

This paper suggests combining LSTMs, trained on a large midi corpus, with a handcrafted reward function that helps to fine-tune the model in a musically meaningful way. The idea to use hand-crafted rewards in such a way is great and seems promising for practical scenarios, where a musician would like to design a set of rules, rather than a set of melodies.

Even though some choices made along the way seem rather ad-hoc and simplistic from a music theoretical perspective, the results sound like an improvement upon the note RNN baseline, but we also don't know how cherry picked these results are. 
I am not convinced that this approach will scale to much more complicated reward functions necessary to compose real music. Maybe LSTMs are the wrong approach altogether if they have so much trouble learning to produce pleasant melodies from such a relatively big corpus of data. Aren't there any alternative differentiable models that are more suitable? What about dilated convolution based approaches?

What I don't like about the paper is that the short melodies are referenced as compositions while being very far from meaningful music, they are not even polyphonic after all. I think it would be great if such papers would be written with the help or feedback of people that have real musical training and are more critical towards these details. 

What I like about the paper is that the authors make an effort to understand what is going on, table 1 is interesting for instance. However, Figure 3 should have included real melody excerpts with the same sound synthesis/sample setup. Besides that, more discussion on the shortcomings of the presented method should be added.

In summary, I do like the paper and idea and I can imagine that such RL based fine-tuning approaches will indeed be useful for musicians. Even though the novelty might be limited, the paper serves as a documentation on how to achieve solid results in practice.

[Author Response · Natasha Jaques · 12 Jan 2017]
**Summary of changes**

The following changes to the paper were made in response to comments:
-Updated related work with additional references
-Updated Section 3.1 to better explain KL control, and to add more explanation on prior work, the history of optimal control, and how reward augmentation can work with SOC
-Re-wrote the abstract and introduction to better reflect the paper’s contributions
-Removed references to composition
-Change Psi-learning to ‘generalized Psi learning’
-Updated Section 6 to add more discussion about the musical shortcomings of the model 
-Modified language in Section 6 to ensure the G model’s different trade-off between data probability and reward was not referred to as worse
-Updated Section 3 to give more detail on how the states and actions work with the recurrent Q-network

[Final Decision · Program Chairs · 06 Feb 2017]
**ICLR committee final decision**

The reviewers generally liked the application; there were a number of technical points raised that leave doubt about the novelty of the approach. However, this may be an interesting avenue in the future, thus the PCs are accepting it to the workshop track.